# Field Experiments of Phyto-Stabilization, Biochar-Stabilization, and Their Coupled Stabilization of Soil Heavy Metal Contamination around a Copper Mine Tailing Site, Inner Mongolia

Hong Liu, Yanguo Teng *, Nengzhan Zheng, Linmei Liu, Weifeng Yue, Yuanzheng Zhai 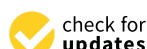 and Jie Yang

College of Water Science, Beijing Normal University, Beijing 100875, China; 201931470037@mail.bnu.edu.cn (H.L.); 201921470034@mail.bnu.edu.cn (N.Z.); 202021470017@mail.bnu.edu.cn (L.L.); yuewf@bnu.edu.cn (W.Y.); zyz@bnu.edu.cn (Y.Z.); yangjie@bnu.edu.cn (J.Y.)
* Correspondence: ygteng@bnu.edu.cn

**Abstract:** A field trial was conducted in Inner Mongolia to evaluate the stabilization effects of phyto-stabilization, biochar-stabilization, and their coupled stabilization for As, Cu, Pb, and Zn in soil. Stabilization plants (*Achnatherum splendens, Puccinellia chinampoensis*, and *Chinese small iris*) and biochar (wood charcoals and chelator-modified biochar) were introduced in the field trial. The acid-extractable fraction and residual fraction of the elements were extracted following a three-stage modified procedure to assess the stabilization effect. The results after 60 days showed that the coupled stabilization produced a better stabilization effect than biochar-/phyto- stabilization alone. *Achnatherum splendens* and *Puccinellia chinampoensis* were found to activate the target elements: the residual fraction proportion of As, Cu, Pb, and Zn decreased while the acid-extractable fraction proportion of Cu and Zn increased in the corresponding planting area. Neither type of biochar produced a notable stabilization effect. The residual fraction proportion of As (20.8–84.0%, 29.2–82%), Pb (31.6–39.3%, 32.1–48.9%), and Zn (30.0–36.2%, 30.1–41.4%) increased, while the acid-extractable fraction proportion remained nearly unchanged after treatment using *Chinese small iris*-straw biochar or *Achnatherum splendens*-straw biochar, respectively. The results indicate that phyto-stabilization or biochar-stabilization alone are not suitable, whereas the coupled stabilization approach is a more efficient choice.

**Keywords:** tailings; heavy metals; biochar-stabilization; phyto-stabilization



## 1. Introduction

China is a major producer of non-ferrous metals, accounting for nearly half of the total global production; However, the production of non-ferrous metals also causes tremendous environmental pollution, and large amounts of tailings are inevitably generated during mine beneficiation [1]. Storage in outdoor yards is one of the most common and extensively used means for tailings disposal because of its low cost. Nevertheless, this approach tends to generate dust containing heavy metals that can be easily spread in the air by wind transport [2], and rain can also introduce these elements into soil and water [3]. Sulfur-containing minerals or alkaline-beneficiation agents in tailings might lead to the acidification [4] or alkalization [5] of soil. The accumulation of toxic elements in the environment surrounding a tailings yard may affect human health [6]. These elements can accumulate in animal bodies, including human beings [7], and heavy metal pollution has been identified to harm neurons, the hematological system, kidneys, bones, and the reproductive system, among other parts of the bodys [8–10]. The relationship between heavy metals and human cancer incidence has been verified [11]. It is therefore of critical importance to take appropriate measures to reduce the environmental risks of tailings

stacks on the surrounding area. Stabilization methods, such as physical (vitrification), chemical (soil washing, solidification and immobilizing agent addition), and biological (phyto-stabilization) approaches, have been applied to reduce the contents or mobility of toxic elements in soil contaminated by tailings [12]. Vitrification has had few site-scale application cases owing to energy consumption limitations. Chemical stabilization is one of the most commonly used stabilization approaches in soil remediation projects. Chelating agents [13], phosphates [14], biochar [15], and red mud [16] have been commonly applied for heavy metal stabilization. Solidified or stabilized heavy metal treatment using a hydraulic binder is also an effective method to reduce leachable heavy metals. Common Portland cement [17] or other binders containing calcium oxide [18] have also been used to stabilize heavy metals in the soil surrounding smelters; However, the agents used in soil washing may result in secondary pollution, and the addition of stabilization/solidification materials may alter the soil ecological function. High costs are also an inevitably limiting factor for physical/chemical stabilization.

Phyto-stabilization has attracted extensive attention because of its eco-friendly and cost-effective characteristics compared with physical/chemical stabilization. Although phyto-stabilization cannot reduce the content of heavy metals in soil, plant roots and associated microorganisms can reduce the mobility and bioavailability of heavy metals in soil [19]. Phyto-stabilization has been applied in some site-stabilization programs, including tailings-containing sites [20], highway adjoining sites [21], and farmland [22]. Hyperaccumulation plants have been used in Shimen (Hunan Province, China) to remediate the soil contaminated by As-bearing tailings [23]. The total stabilization area was 68 hm$^2$ and the As content in soil was reduced by 13.6% after two years of stabilization.

Biochar has been considered to be a potential soil remediation material in recent years owing to its low cost, low toxicity, and abundant resources. Biochar materials have a very large specific area, porous structure, and abundant surface oxygen functional groups, and can therefore easily combine with heavy metals in soil. Aside from adsorption, biochar can also react with toxic elements through electron transfer via its aromatic and graphene structures. A typical example is the reduction transformation of Cr(VI) to Cr(III) [24]. Chelation reactions and chemical precipitation also occur during the biochar stabilization process. Although biochar can combine with heavy metal cations in soil [25], its stabilization effects on arsenic (As) are limited [26]. Modified biochar has thus been produced to fulfill different demands. Common modified approaches include iron-based or magnetic modification [27,28], acid or alkali modification [29], and chitosan modification [30]. In addition to its use in pollution removal, biochar can also provide potential agronomic benefits [31]. The addition of biochar can change the pH and nutrient element distribution in soil [32]. Biochar can also alter the soil microorganism community's composition and activity, which is important for root development and nutrition transport, and may thus promote plant growth [33]. Long stabilization periods have limited the application of phyto-stabilization, whereas the introduction of biochar can shorten the stabilization cycle. Some studies have shown that biochar can reduce the bioavailability of heavy metals in soil, but does not produce any effect or enhancement of heavy metal enrichment of the hyperaccumulator [34]. The combination of phyto-stabilization and biochar stabilization may therefore be a better alternative compared with traditional single-method approaches. A series of biochar-based soil stabilization projects have been conducted on tailings-contaminated soil. For example, the application of biochar notably reduced the extractable fraction of Pb, Zn, and Cu [35] in the soil of a smelter in Shanghai China.

The major non-ferrous metal production bases in China are situated in several provinces, including Yunnan, Guizhou, Jiangxi and Inner Mongolia, where soil pollution incidents caused by heavy metals are frequent. Although there had been large number of study focused on soil pollution in the eastern and southern regions of China, few studies focused on soil pollution in Inner Mongolia, one of the main non-ferrous metal production bases in the country. Similar to the north and west of Inner Mongolia, the climate in this area is dry and the soil properties differ from those in southern regions of China. A site-scale

stabilization test is important for applying existing technology in Inner Mongolia. In this study, a pasture that was formerly a tailings storage yard was chosen as the trial site. Three stabilization techniques were applied to reduce the mobility of As, Cu, Pb, and Zn in the onsite soil, including phyto-stabilization, biochar stabilization, and a coupled stabilization of these two stabilization means. The proportion of As, Cu, Pb, and Zn divided by a three-stage modified procedure recommended by BCR (European Community Bureau of Reference) was determined before and after the trial. The change of acid-extractable fraction proportion and residual fraction proportion were used to assess the stabilization effect of the selected techniques. The aims of this work were as follows: (i) to verify the stabilization effect of common technology through field scale trials, rather than batch trials in the laboratory; and (ii) to propose the suitable stabilization mean under the environment of studied area.

## 2. Materials and Methods

### 2.1. Study Area

The study area was situated in an abandoned weather tailings yard of a copper smelter in Bayannur, Inner Mongolia (106.60° E, 41.28° N). The area now is a pasture. The region was located in a mid-temperate zone with an annual average temperature of 3.9 °C. The area is prone to drought with annual rainfall of 96–105.9 mm, and the extent of evaporation is approximately 14 times that of precipitation [36]. The tailing stack is located in the south of the study area. No pollution control means were applied to these weather tailings stacks, thus, the surrounding area was polluted by toxic elements including Cu, Pb, As, and Zn. The polluted surrounding area was defined as the so-called "tailing stacks effected area", and the trial was conducted in the north of this area. The investigation showed high Cu and Pb contamination levels and relatively low Zn and As contamination levels. The average contents of Cu, Pb, As, and Zn in the trial site were 716, 263, 30.2, and 145 mg·kg$^{-1}$, respectively. The trial site was divided into three parts for three stabilization techniques (phyto-stabilization, biochar-stabilization, and coupled stabilization, i.e., a combination of both stabilization methods), and the corresponding stabilization effects were examined. The location of the trial field and field planning is shown in Figure 1.

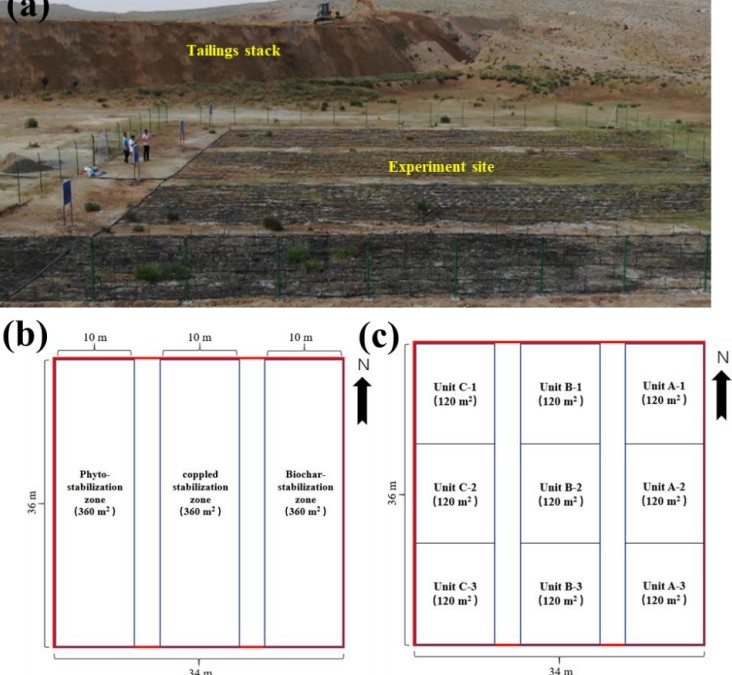

**Figure 1.** Location of the trial site and site division: (**a**) panoramic view of experiment site; (**b**) division of different stabilization zones in the trial site; and (**c**) distribution of sampling units.

### 2.2. Phyto-Stabilization Experiment

Typical stabilization plants, such as *Pteris vittatal* [37] and *Sedum alfredii Hance* [38], were not introduced owing to the arid climate and salinization of the soil in the study area. Instead, native plants of the region, including *Phragmites*, *Achnatherum splendens*, and *Chinese small iris*, were investigated. The content of As, Cu, Pb, and Zn in the aerial parts and roots of these plants are listed in Table 1.

**Table 1.** Heavy metal contents in the aerial parts and roots of native plants in the study area.

|  | **Plant Species** | **As (ppm)** | **Cu (ppm)** | **Pb (ppm)** | **Zn (ppm)** |
|---|---|---|---|---|---|
| aerial part | *Phragmites* | 0 | 59.12 | 10.81 | 42.68 |
| | *Achnatherum splendens* | 0.70 | 218.53 | 22.65 | 72.82 |
| | *Chinese small iris* | 2.66 | 85.22 | 31.37 | 39.61 |
| root | *Phragmites* | 2.90 | 110.70 | 45.59 | 34.46 |
| | *Achnatherum splendens* | 0.22 | 721.67 | 36.94 | 147.45 |
| | *Chinese small iris* | 0 | 44.66 | 19.12 | 35.83 |

*Achnatherum splendens* and *Chinese small iris* showed a relatively higher heavy metal content in the aerial part. The heavy metal content in the *Phragmites*' biomass was considerably lower than that of the other two investigated plants. Accordingly, *Achnatherum splendens* and *Chinese small iris* were chosen as the native stabilization plants. A type of salinity tolerant forage in Inner Mongolia, *Puccinellia chinampoensis* [39], was also introduced into the trial. All three kinds of plants were planted in the phyto-stabilization zone with equal planting areas (120 m$^2$). Each seedling was planted with a row spacing of 1 m and plant spacing of 0.5 m (Figure 2). Suitable holes were excavated before planting, and seedlings were then placed into the holes.

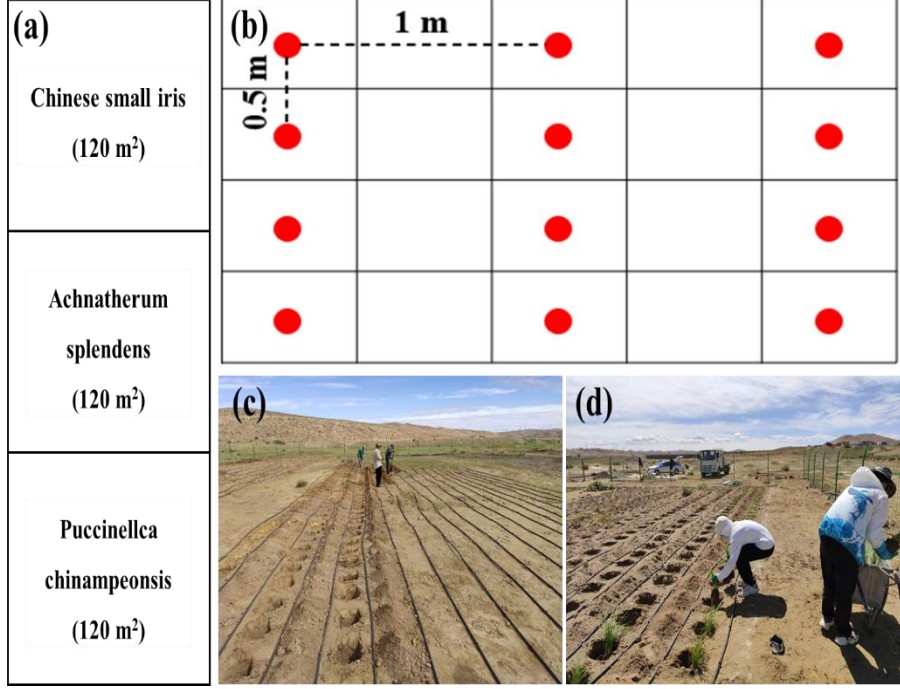

**Figure 2.** Phyto-stabilization zone and planting: (**a**) plot of planting in the phyto-stabilization zone, (**b**) row spacing and plant spacing, (**c**) planting hole excavating, (**d**) seedling planting and soil backfilling.

### 2.3. Biochar-Stabilization Experiment

Prior to adding biochar, the top soil (0–20 cm [40]) in the zone was dug and cultivated. The biochar was then added into the soil and artificially mixed with the soil at a dosage of 5% (*w/w*) [41]. The zone was divided into square blocks of equal area, and a bag of biochar

was placed into each block in advance. Each bag of biochar was opened individually and thoroughly mixed with the soil in the block. The process was repeated in all blocks, one block a time. The workflow was shown in Figure 3.

Two kinds of biochar were chosen: wood charcoals and chelator-modified straw biochar. The wood charcoal was purchased from Senqi Environmental Technical Co., Ltd. in Shijiazhuang, China. The modified biochar was produced through an adsorption procedure. Corn stalk biochar was added into a solution of chelating agent with a solid–liquid ratio of 0.1 g/mL and agitated for 24 h. The solid was then separated and washed with deionized water several times until the water pH was equal to 7. The solid was heated at 60 °C for 12 h to produce the final modified biochar product.

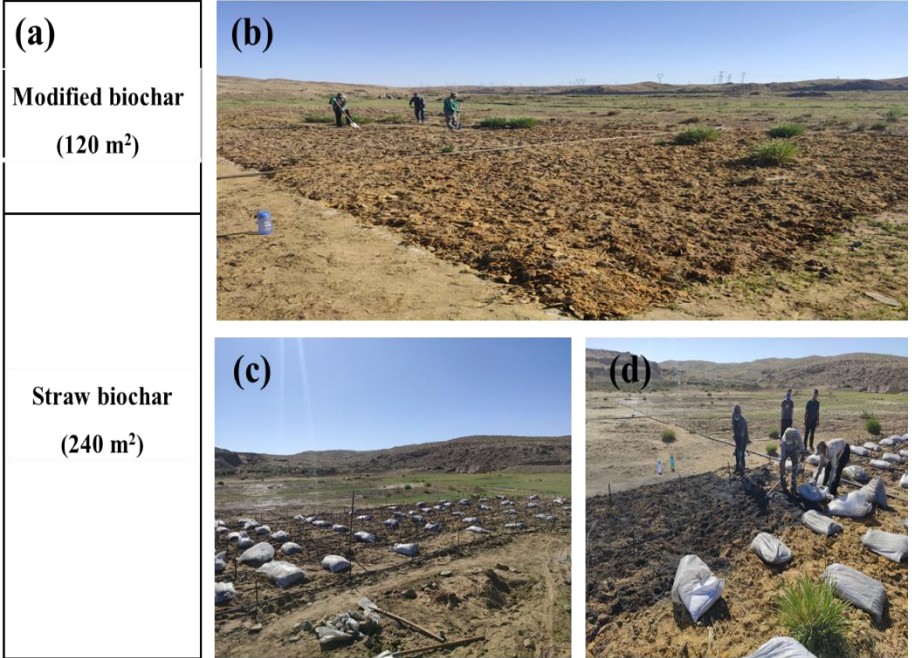

**Figure 3.** Biochar-stabilization and biochar application: (**a**) plot of the biochar-stabilization zone, (**b**) cultivation of the soil in the biochar-stabilization zone, (**c**) transport of biochar, (**d**) artificial mixing of biochar, and top soil.

### 2.4. Coupled Stabilization Experiment

In the coupled stabilization zone, both the phyto-stabilization and biochar-stabilization techniques mentioned above were applied simultaneously. The plant species, row spacing, and plant spacing were as same as those of the phyto-stabilization zone. The wood charcoals used in biochar-stabilization were also added into soil at the same dosage of 5% (*w/w*). Seedlings were placed into the pre-dug holes and a mixture of soil and biochar was backfilled. The regional division and workflow are shown in Figure 4.

### 2.5. Sample Collection, Handling, and Analyses

Each zone of the stabilization experiment was divided into three quadrilateral sampling units with an area of 120 m². To avoid soil variation, the sampling points were settled on the quarter points and cross point of the diagonals in each unit. Equiponderant soil was collected from each sampling point in the units that were mixed, and the mixed samples were used to reflect the pollutant situation of the corresponding units. Soil samples were collected at the beginning of the trial test and 30 and 60 days thereafter.

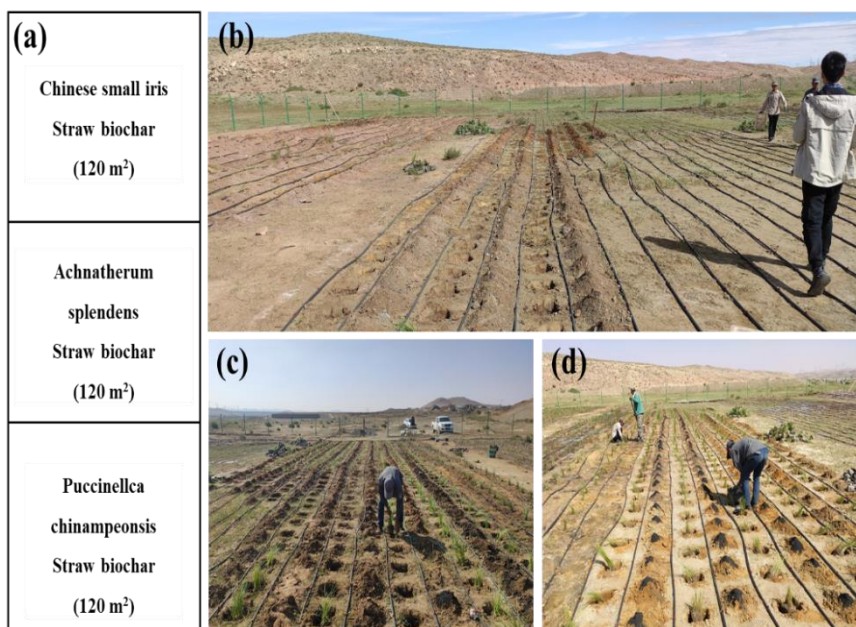

**Figure 4.** Coupled stabilization zone: (**a**) plot of the complex stabilization zone, (**b**) planting holes excavating, (**c**) mixing of biochar and soil, and (**d**) seedling planting and soil backfilling.

Soil collected from all points was dried in air and sieved through a 100-mesh sieve [42] to remove pebbles or other materials. Equiponderant treated soil from five points in one sampling unit were thoroughly mixed. The mixed sample was used to analyze and reflect the property of the corresponding sampling unit. The total content of Cu, Pb, As, and Zn in soil were determined using an axial view inductively coupled plasma spectrometer (SPECTRO ARCOS, SPECTRO Analytical Instruments, Kleve, Germany). A 0.1 g sample of mixed soil was put into Poly tetra fluoroethylene tubes, and 5 mL of $HNO_3$, 2 mL of HF, and 1 mL of $HClO_4$ were successively added into each tube [43]. The soil and mixed acid were heated and the obtained solution was prepared up to 20 mL, with distilled water. All solution samples were filtered through a 0.22-μm membrane for subsequent analysis.

The fractions of each element in the soil were extracted via a three-stage modified sequential extraction procedure [44]. According to the order of extraction, the total content of elements was divided into the acid-extractable fraction, reducible fraction, oxidizable fraction, and residual fraction. The procedure is described as follows. Place 1 g of the sieved mixed sample into the tubes, then add 20 mL of acetic acid solution (0.11 M). Sixteen hours of shaking was applied at room temperature. After being shaken, the residual solid was separated via centrifugation (4000 rpm, 20 min), and the supernatant was collected for acid-extractable fraction analysis. Twenty milliliters of hydroxylamine hydrochloride were then added to the solid, washed with distilled water following the same extraction and centrifugation procedure, and the supernatant was collected for reducible fraction analysis. Five milliliters of hydrogen peroxide (30%) were used to oxidize the sample (evaporated hydrogen peroxide in an 85 °C bath to approximately 1 mL), followed by the addition of ammonium acetate (1 M, pH = 2.0) to extract the oxidizable fraction using the same procedure. The residual solid was digested as the process used the total content analysis to quantify the residual fraction of each element.

The soil pH was measured as follows. Ten grams of dried and sieved soil was weighed and put into a flask. After adding 25 mL of deionized water, the mixture of water and soil was vigorously shaken for 5 min and then allowed to stand for 2 h. The pH of the supernatant was measured using a pH meter as the soil pH.

### 3. Results

#### 3.1. Heavy Metal Fractionation in the Phyto-Stabilization Experiment

The proportion of different fractions of As, Cu, Pb, and Zn before and after the phyto-stabilization application is shown in Figure 5. In the phyto-stabilization zone, the acid-extractable fraction of As and Pb in the original soil was low. More than 80% of As belongs to the residual fraction, while the percentages of residual fraction Pb in the three sampling units of the phyto-stabilization zone were 62%, 88%, and 79%. Because the residual fraction is considered nonbioavailable [45], the relatively high proportion of the residual fraction indicates that As and Pb were chemically stable in the phyto-stabilization zone. Correspondingly, almost no acid-extractable fraction of As and Pb was detected in the original soil, which implies that As and Pb in the phyto-stabilization zone were unleachable in rainfall erosion. Compared with As and Pb, Zn and Cu showed higher mobility and a higher proportion of acid-extractable fractions in the original soil. The proportions of acid-extractable fraction Cu in the original soil were 29.6%, 4.6%, and 5.8% for unit C-1, unit C-2, and unit C-3. For Zn, the proportions were 45.0%, 10.1%, and 5.2%, respectively. After 60 days of stabilization, the proportion of the residual fraction of As in all units decreased, and Cu, Pb, and Zn showed the same trend. In units C-2 and C-3, the residual fraction of Cu disappeared after 60 days of stabilization, while the proportion of the residual fraction of Pb decreased from 88.7% to 37.3% in C-2, and 79.2% to 22.5% in C-3. The acid-extractable As, Zn, and Cu increased after 60 days of stabilization. For Pb and Cu, the reducible fraction notably increased and the residual fraction decreased. This phenomenon revealed a potential transformation from one residual fraction to the other fraction under the phyto-stabilization application.

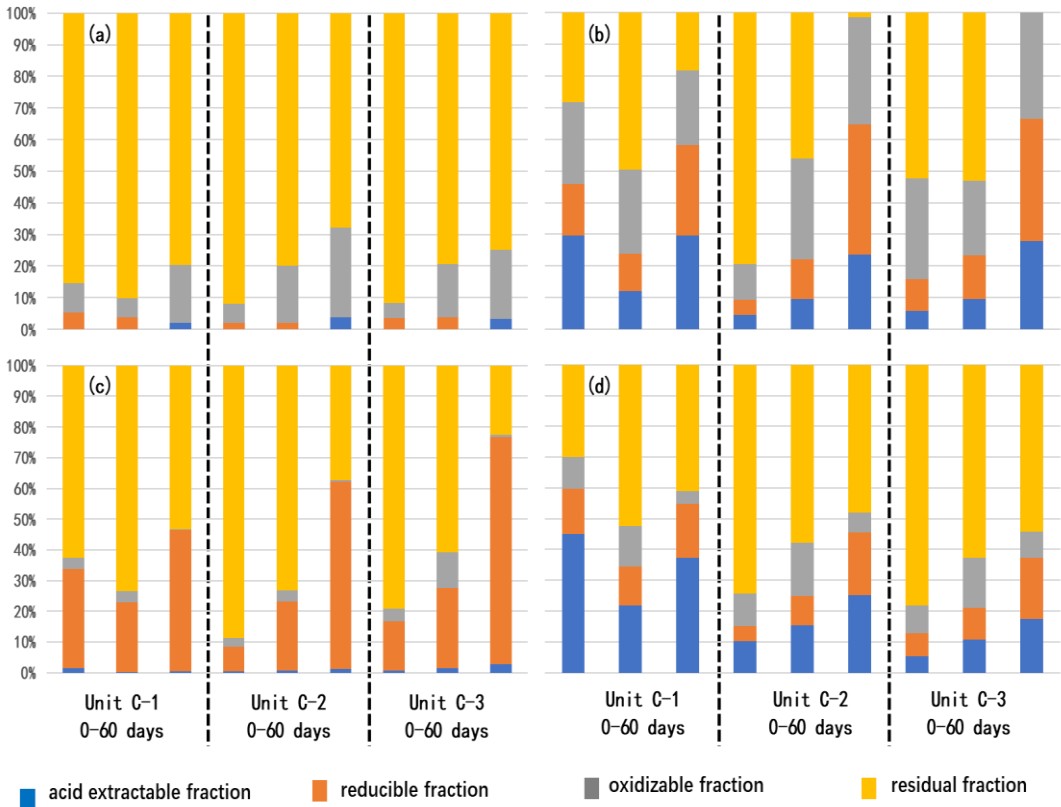

**Figure 5.** Effect of phyto-stabilization after 30 and 60 days: (**a**) fraction distribution of As after 0, 30, and 60 days; (**b**) fraction distribution of Cu after 0, 30, and 60 days; (**c**) fraction distribution of Pb after 0, 30, and 60 days; (**d**) fraction distribution of Zn after 0, 30, and 60 days.

### 3.2. Heavy Metal Fractionation in the Biochar-Stabilization Experiment

The stabilization effect of biochar-stabilization is shown in Figure 6. In the biochar-stabilization zone, all sampling units showed a higher proportion of acid-extractable fractions of As and Pb than that of the phyto-stabilization zone. Aside from unit C-1, the samples from the other two units also had a higher proportion of acid-extractable Cu and Zn. After 60 days of stabilization, the As in two of the units showed a decline in the residual fraction, while acid-extractable fraction showed no notable change. The proportion of acid-extractable Cu fluctuated during stabilization, and the proportion of the residual fraction decreased in units A-1 and A-3. The proportion of acid-extractable Zn initially decreased and then increased, corresponding to the reduction of residual Zn in all units. In units A-1 and A-2, the residual fraction of Pb decreased, and the proportion of reducible fraction substantially increased. As in the phyto-stabilization zone, the environmental risk caused by toxic elements was not reduced after stabilization. This result was unexpected because biochar is considered to be an effective remediator for heavy metals based on its high absorption capacity in lab-scale tests [46] and site stabilization.

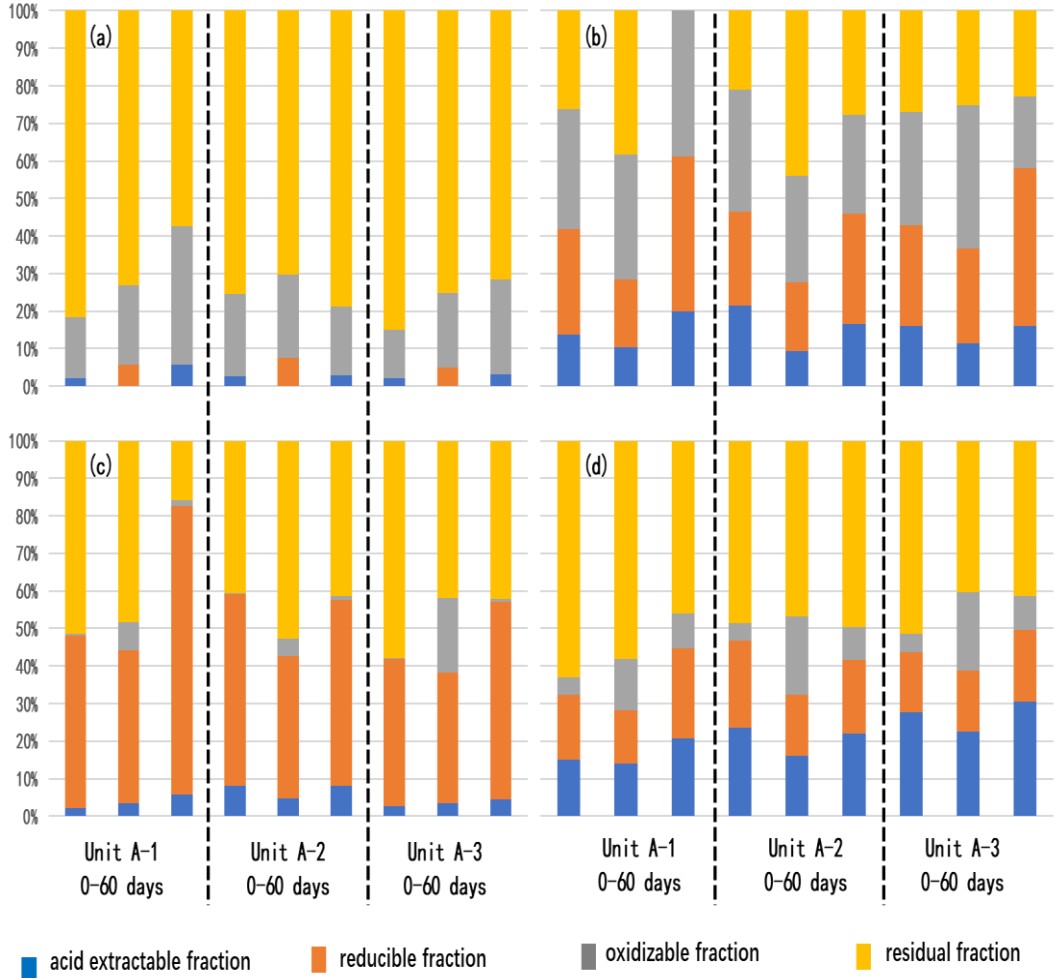

**Figure 6.** Effect of biochar-stabilization after 30 and 60 days: (**a**) the fraction distribution of As after 0 days, 30 days and 60 days; (**b**) the fraction distribution of Cu after 0 days, 30 days, and 60 days; (**c**) the fraction distribution of Pb after 0 days, 30 days and 60 days; and (**d**) the fraction distribution of Zn after 0 days, 30 days, and 60 days.

### 3.3. Heavy Metal Fractionation in the Coupled Stabilization Experiment

As shown in Figure 7, the residual fraction proportion of As in the native soil was substantially less than that in the other zones. Although the content of acid-extractable

As increased, the proportion of residual fraction increased to ~80% in unit B-1 and unit B-2 after 60 days, indicating an outstanding stabilization effect. The increased residual fraction proportion was also observed for Pb in units B-1 and B-2 with the acid extractable fraction proportion having decreased from 6.8% to 4.0% in unit B-1, and from 6.8% to 2.9% in unit B-2; However, the coupled stabilization didn't show a similar stabilization effect for Cu. The acid-extractable Cu proportion increased after stabilization with the residual fraction proportion decreasing gradually. In units B-1 and B-3, the residual fraction, Cu, disappeared after the stabilization experiment with the acid extractable fraction, the Cu proportion, increased to 56.5%, 54.7%, and 44.6% in unit B-1, unit B-2, and unit B-3, respectively. The proportion of acid-extractable Zn increased in all units with stabilization; However, the residual fraction proportion in units B-1 and B-2 also increased. Compared with phyto-stabilization and biochar-stabilization, the coupled stabilization produced a better stabilizer effect with a higher residual fraction proportion of As, Cu, and Zn after the stabilization experiment, which was not observed in the other stabilization zones.

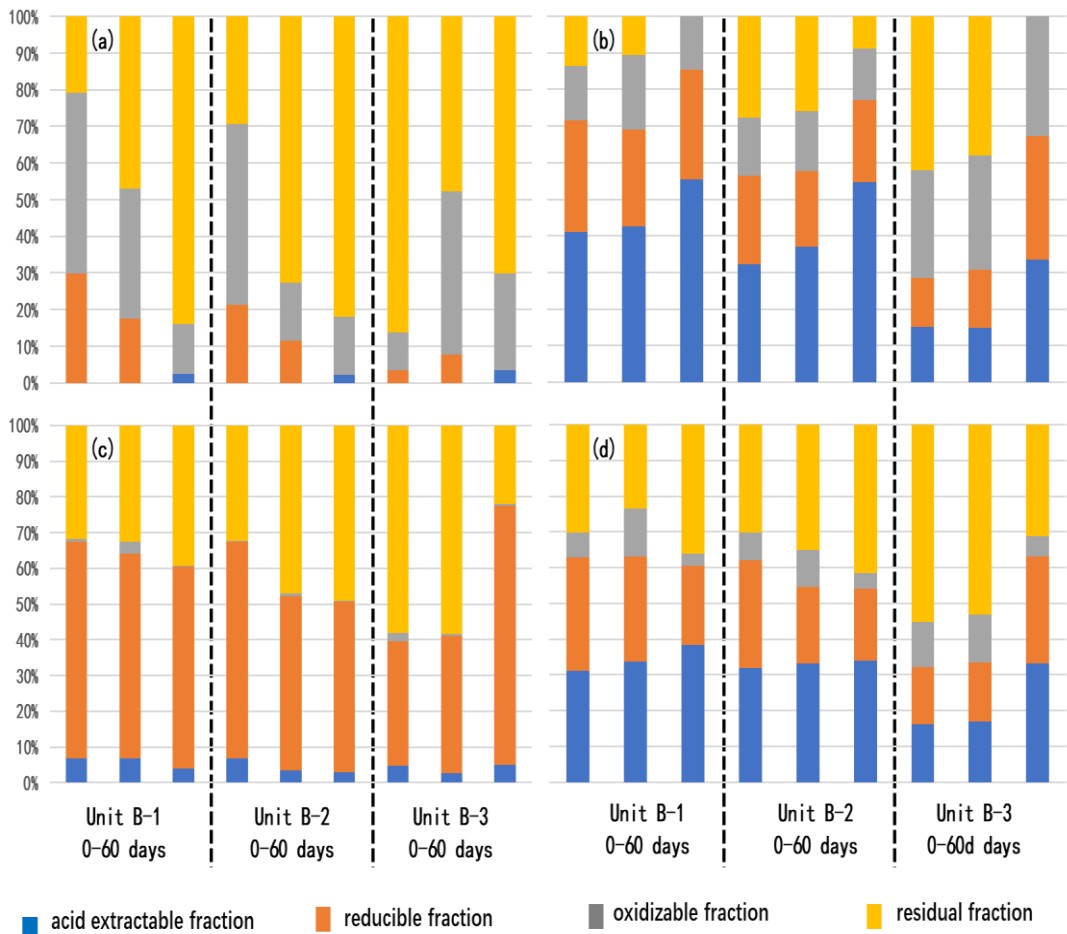

**Figure 7.** Effect of coupled stabilization after 30 and 60 days: (**a**) fraction distribution of As after 0, 30, and 60 days; (**b**) fraction distribution of Cu after 0, 30, and 60 days; (**c**) fraction distribution of Pb after 0, 30, and 60 days; (**d**) fraction distribution of Zn after 0, 30, and 60 days.

## 4. Discussion

### 4.1. Evaluation of Different Stabilization Means

In phyto-stabilization zone, especially in units C-2 and C-3, the residual fraction proportion of all elements decreased, and the acid extractable fraction proportion of Cu and Zn increased obviously after the experiment, indicating that phyto-stabilization didn't get the stabilization effect and even activate the Cu and Zn in the zone. In other words, phyto-stabilization did not reduce the environmental risk of As, Cu, Pb, or Zn. *Achnatherum*

*splendens* and *Puccinellia chinampoensis* might not be suitable for phyto-stabilization. A possible explanation for this is that the plant root exudates interact with heavy metal elements in the soil, thus enhancing their solubility and resulting in the transformation of the residual fraction [47]. The organic acids in root exudates might react with heavy metals in the soil, and thus improve the mobility of heavy metals through chelation [48]. According to a weathering experiment by Anna Potysz et al., the artificial root exudates improve the release of elements from Zn–Pb-bearing rocks [49], which showed the possibility of heavy metal activation through root exudates.

Reduction of residual fraction proportion and the increase of an acid extractable fraction of elements were also observed in the biochar-stabilization zone, which showed that biochar-stabilization wasn't an effective stabilization mean in this experiment. Some potential reasons leading to the poor stabilization effect might be as follows: (i) The first reason might be the low content of acid fraction in the original soil. The original proportion of the acid-extractable fraction was lower than 10% for As and Pb, and lower than 20% for Cu. The high chemical activity of the acid-extractable fraction makes it more easily react with the biochar surface. (ii) The second reason is the arid climate and water scarcity. Related studies have shown that the soil treated with biochar consistently maintained a relative stable moisture content (70% or higher) during the entirety of the stabilization trials [50–52]. The annual precipitation in this area is only ~100 mm and the original soil in stabilization zone was dry. Restricted by limited water resources, the soil was irrigated every five days to preserve plant growth, but this amount was insufficient to keep the soil always consistently wet. (iii) The third reason might be limitations of pristine biochar. As a kind of alkaline material, pristine biochar can bind many cations; However, the application of biochar might improve the bioavailability of As in soil [53,54]. The application of biochar would generate dissolved organic carbon (DOC). The DOC could compete with As for adsorption onto minerals in soil. This could explain the decrease of the proportion of residual fraction As in the biochar-stabilization zone.

Compared with phyto-stabilization and biochar-stabilization, the application of coupled stabilization increased the residual fraction proportion of As, Pb, and Zn, and behaved a better stabilization effect. Besides B-3, the residual fraction proportion of As, Pb, and Zn, in units B-1 and B-2, increased after the experiment. This demonstrates that the combination of *Chinese small iris*-straw biochar and *Achnatherum splendens*-straw biochar had a better stabilization ability for heavy metals in the soil than that of the *Puccinellia chinampoensis*-straw biochar. The relatively higher non-residual fraction proportion of heavy metals in the original soil might be a reason. The results could also be attributed to plant exudate activation. Previous studies have shown that exudates, including organic acids and biosurfactants, can increase the pH value of soil and the mobility of heavy metal elements [55]. The higher acid-extractable fraction proportion in the phyto-stabilization zone and coupled stabilization zone verifies the activation effect of the stabilization plants. The increased mobility promoted the combination of biochar and heavy metal elements. This process can explain the poor stabilization effect in the biochar-stabilization zone with a high residual fraction proportion in the native soil, and why the acid-extractable fraction proportion increased in the phyto-stabilization zones.

### 4.2. Relationship between Change of Fraction

The pH is a critical parameter for heavy metal stabilization; the soil pH of all sampling units was therefore measured before and after the stabilization treatment (Table 2). The original soil pH of the trial site was neutral, except for unit C-1. After 60 days of treatment, the soil pH in the biochar-stabilization zone remained constant, that in unit B-2 decreased, and that in the phyto-stabilization zone substantially increased.

The Pearson correlation coefficients between pH, acid-extractable fraction, and residual fraction were calculated (Figure 8) to determine the effect of the pH change on the heavy metal fraction and identify the potential relationship between the change of heavy metal fraction. The soil pH showed an overall positive correlation with the acid-extractable

fraction of the heavy metals and an overall negative correlation with the residual fraction; However, the correlation was not significant, indicating that the change of soil pH was not the main mechanism to achieve the stabilization effect of the chosen technique. In the coupled stabilization zone and phyto-stabilization zone, there was a significant negative correlation between the acid-extractable fraction proportion and the residual fraction proportion, which implies a potential transformation between the two fractions, while a significant correlation was not observed in the biochar stabilization zone. The relative positive correlation between the residual fraction proportion of Zn, Cu, and Pb, and the negative correlation between their residual fraction and acid-extractable fraction, demonstrate the analogous stabilization process and mechanism of these elements.

**Table 2.** Soil pH of all sampling units at 0, 30, and 60 days.

| Sampling Date | Biochar-Stabilization | | | Coupled Stabilization | | | Phyto-Stabilization | | |
|---|---|---|---|---|---|---|---|---|---|
| | Unit A-1 | Unit A-2 | Unit A-3 | Unit B-1 | Unit B-2 | Unit B-3 | Unit C-1 | Unit C-2 | Unit C-3 |
| 0 days | 7.69 | 7.67 | 7.15 | 7.42 | 7.33 | 7.47 | 5.5 | 6.76 | 7.01 |
| 30 days | 7.4 | 7.45 | 7.44 | 7.4 | 6.34 | 7.19 | 7.3 | 7.45 | 7.64 |
| 60 days | 7.71 | 7.63 | 7.54 | 7.56 | 6.73 | 7.44 | 7.08 | 7.37 | 7.44 |

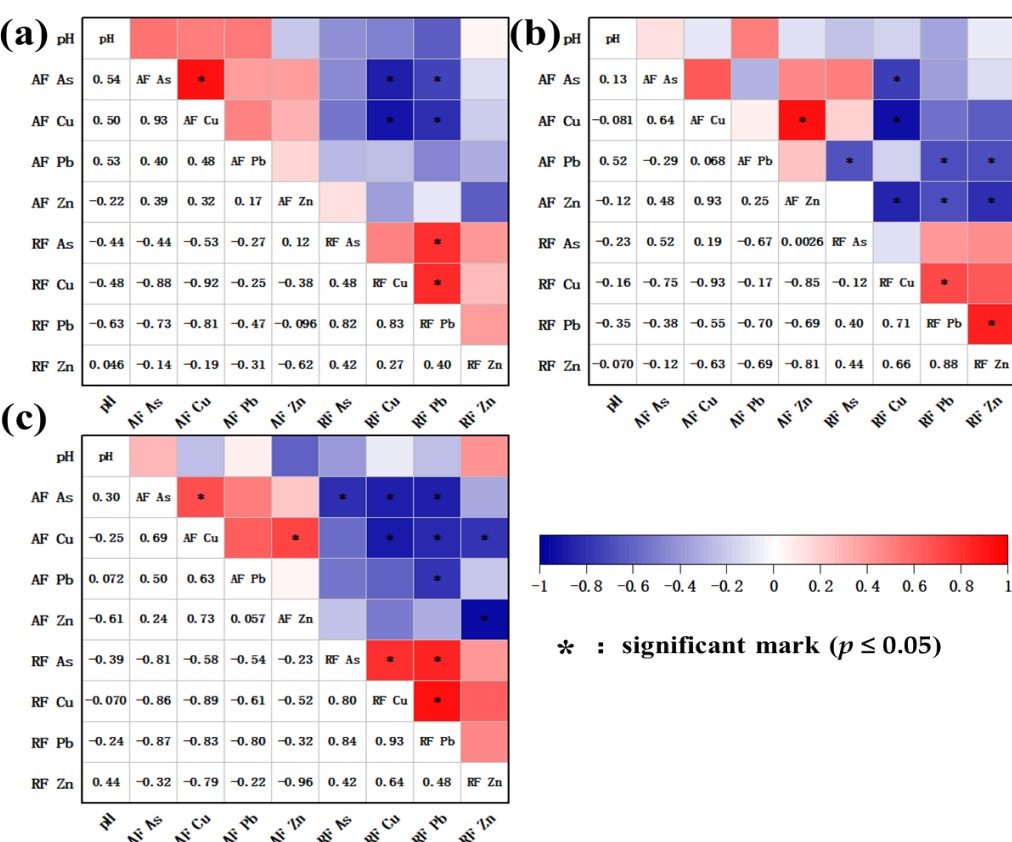

**Figure 8.** Correlation matrix of the pH and element fraction in the (**a**) biochar-stabilization zone, (**b**) complex stabilization zone; (**c**) phyto-stabilization zone.

## 5. Conclusions

The tests reveal that phyto-stabilization did not produce a stabilization effect, because the residual fraction proportion of As, Cu, Pb, and Zn decreased after stabilization, and the acid-extractable As and Zn increased. Some units also showed a higher acid-extractable proportion of Cu and Zn. This might be attributed to the activation of root secretion. A similar decrease of the residual fraction proportion of all elements was observed in the

biochar-stabilization zone. This demonstrates that a single approach of phyto-stabilization or biochar-stabilization cannot stabilize heavy metals in the study area. In the coupled stabilization zone, the residual fraction proportion of As, Pb, and Zn, in units B-1 and B-2, increased after stabilization, thus showing the higher passivation effects of coupled stabilization than in a single approach. The activation-stabilization mechanism of the plant–biochar system might explain the improved stabilization effect. In summary, a single stabilization approach might not be a suitable scheme, whereas the combination of phyto-stabilization and biochar-stabilization may be a better choice in site stabilization practice.

**Author Contributions:** H.L.: Formal analysis, Investigation, Writing—original draft. Y.T.: Conceptualization, Supervision, Methodology. N.Z.: Field investigation. L.L.: Heavy metal analysis. W.Y.: Soil contaminated assessment. Y.Z.: Soil monitoring. J.Y.: Extracted heavy metal in the soil. All authors have read and agreed to the published version of the manuscript.

**Funding:** This research was supported by the Key Science and Technology Projects of Inner Mongolia Autonomous Region (2019ZD001-05) and Beijing Advanced Innovation Program for Land Surface Science.

**Data Availability Statement:** Not applicable.

**Acknowledgments:** The authors are grateful to the support and help from the Analytical and Testing Center of Beijing Normal University.

**Conflicts of Interest:** The authors declare that they have no known competing financial interests or personal relationships that could have appeared to influence the work reported in this paper.

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
