# Peer review of "Field Experiments of Phyto-Stabilization, Biochar-Stabilization, and Their Coupled Stabilization of Soil Heavy Metal Contamination around a Copper Mine Tailing Site, Inner Mongolia"

_minerals, doi:10.3390/min12060702_

Round 1
Reviewer 1 Report
The lack of line numbers makes it difficult to complete a review.
1. Introduction - The chapter should end with the aim of the work.
2.1 Study area: "temperature of 3.9 ° C." - Please remove the space before the "°".
2.1: "shown in Figure 1" - The sentence should end with ".".
2.1: "Figure 1." - I suggest to improve the quality of the figure in the final version.
2.5: "temperature of 85 ° C" - Please remove the spaces before "°".
Author Response
Point 1: 1. Introduction - The chapter should end with the aim of the work.
Response 1: The aim of the work had been added into the end of introduction, and the new end sentence of introduction was as follows: The aims of this work were as follows: (i) to verify the stabilization effect of common technology through field scale trials, rather than batch trials in the laboratory; (ii) to propose the suitable stabilization mean under the environment of studied area.
Point 2: 2.1 Study area: "temperature of 3.9 ° C." - Please remove the space before the "°".
Response 2: The space before the "°" had been removed.
Point 3: 2.1: "shown in Figure 1" - The sentence should end with ".".
Response 3: The "." had been added to the end of the sentence.
Point 4: 2.1: "Figure 1." - I suggest to improve the quality of the figure in the final version.
Response 4: The figure had been replaced with a new one with higher quality.
Point 5: "temperature of 85 ° C" - Please remove the spaces before "°".
Response 5: The space before the "°" had been removed.

Reviewer 2 Report
The manuscript minerals-1748667 “Field experiments of phyto-stabilization, biochar-stabilization, and their coupled stabilization of soil heavy metal contamination around a copper mine tailing site, Inner Mongolia” are fundamental and applied study of the mine tailing site. The undoubted merit of the authors is the routine and detailed analysis of soil heavy metal contamination based on field experiments. The manuscript is generally well-written, well figured, and interesting. Overall, I would recommend its publication in Minerals. I like this manuscript and think the authors have presented a comprehensive and compelling case which strengthens the available data and interpretations of the phyto- and biochar-stabilization and their coupled stabilization of soil heavy metal contamination. However, there are a few areas where I think the authors can strengthen the manuscript. These are not primarily related to science, but the presentation of their work in the text and the clarity of the findings they are trying to convey. I view these changes as moderate as they will not require substantial revision but careful tweaking and extending a few sections of the manuscript. My main comments relate to the accessibility of the manuscript (i.e., how standalone it is) and the contents of the Discussion section, which I believe require some moderate improvements. These issues are not meant to disparage what is overall a well-written manuscript. Instead, these comments and suggestions are meant to strengthen the use of this data and its potential interest, relevance, and appeal to the international audience Minerals attracts.
Major comments
1. It is recommended to separate the results and their discussions. This will strengthen the manuscript.
2. The main conclusions of this study should be more discussed by exemplifying similar studies from other parts of the world. In other words, the authors should justify their conclusions by comparing their results with others’ works, with more emphasis on the similarities and differences of their findings with those works. In general, this is a problem present throughout the discussion text. The authors should give more credit to others’ works and discuss their findings in the context of previous work. A moderate revision to the discussion is thus recommended.
Minor comments
3. "BCR" to decipher the abbreviation at the first mention.
4. Table 1. Specify the units of measurement of metal concentrations in the header or top line of the table.
5. It is also recommended to do detailed proofreading of the text.
Author Response
Dear Reviewer:
Thank you for your valuable comments and these comments are helpful for improving our paper. The response to the comments are listed as follow:
Point 1: It is recommended to separate the results and their discussions. This will strengthen the manuscript.
Response 1: The results and their discussions had been separated. A new “Discussion” chapter had been added to evaluate the stabilization effect and the poteantial reason had been discussion to explain the fraction change of As, Cu, Pb, Zn after experiments.
Point 2: The main conclusions of this study should be more discussed by exemplifying similar studies from other parts of the world. In other words, the authors should justify their conclusions by comparing their results with others’ works, with more emphasis on the similarities and differences of their findings with those works. In general, this is a problem present throughout the discussion text. The authors should give more credit to others’ works and discuss their findings in the context of previous work. A moderate revision to the discussion is thus recommended.
Response 2: Some related studies had been added in the discussion. In chapter 4.1, the studies about bioweathering experiments and some related examples (reference 48 and reference 49) were introduced, to showed the similar improvement in heavy metals mobility caused by organic acid in root exudates. The added references (reference 53 and reference 54) also revealed the increase of extractable As proportion after biochar application. According to the references, the dissolved organic carbon generated from biochar might lead to the activation of As in soil of biochar-stabilization zone.
Point 3: "BCR" to decipher the abbreviation at the first mention.
Response 3: The interpretation of abbreviation “BCR” had been added.
Point 4: Table 1. Specify the units of measurement of metal concentrations in the header or top line of the table.
Response 4: The units of measurement of metal concentrations (ppm) had been added in the top line of Table 1.
Point 5: It is also recommended to do detailed proofreading of the text.
Response 5: Other details of spelling, units and punctuation had been detailed.